# A cfDNA methylation-based tissue-of-origin classifier for cancers of unknown primary

Alicia-Marie Conway [1,2,5], Simon P. Pearce [3,5], Alexandra Clipson [1,5], Steven M. Hill [3], Francesca Chemi[1], Dan Slane-Tan[1], Saba Ferdous[3], A. S. Md Mukarram Hossain [3], Katarzyna Kamieniecka[3], Daniel J. White[1], Claire Mitchell[4], Alastair Kerr [3], Matthew G. Krebs[2], Gerard Brady[1], Caroline Dive [1,3] ✉, Natalie Cook [2] ✉ & Dominic G. Rothwell [1] ✉

Cancers of Unknown Primary (CUP) remains a diagnostic and therapeutic challenge due to biological heterogeneity and poor responses to standard chemotherapy. Predicting tissue-of-origin (TOO) molecularly could help refine this diagnosis, with tissue acquisition barriers mitigated via liquid biopsies. However, TOO liquid biopsies are unexplored in CUP cohorts. Here we describe CUPiD, a machine learning classifier for accurate TOO predictions across 29 tumour classes using circulating cell-free DNA (cfDNA) methylation patterns. We tested CUPiD on 143 cfDNA samples from patients with 13 cancer types alongside 27 non-cancer controls, with overall sensitivity of 84.6% and TOO accuracy of 96.8%. In an additional cohort of 41 patients with CUP CUPiD predictions were made in 32/41 (78.0%) cases, with 88.5% of the predictions clinically consistent with a subsequent or suspected primary tumour diagnosis, when available (23/26 patients). Combining CUPiD with cfDNA mutation data demonstrated potential diagnosis re-classification and/or treatment change in this hard-to-treat cancer group.

The diagnostic and therapeutic challenges in Cancers of Unknown Primary (CUP) are an increasingly important unmet clinical need, recently exemplified by several large studies and updated ESMO CUP guidelines[1–4]. Improved management pathways attempt to aid diagnosis of otherwise 'diagnostically challenging' primary tumours often misclassified as CUP[3]. These pathways also seek to simplify and expand classification of 'favourable' CUP subsets defined as tumours clinically aligned to known tumour types[3]. Patients within the 'favourable' subset (~20%) can access the aligned tumour site-directed therapies and achieve better clinical outcomes[5]. However, for these patients arriving at a primary tumour diagnosis usually involves a lengthy diagnostic journey, comprising multiple investigations and pathology reviews, from a very limited quantity of tissue obtained via an invasive tumour biopsy.

For patients within the 'unfavourable' CUP subset, treatment is limited to 'one-size-fits-all' chemotherapy despite a wealth of clinical, pathological and molecular heterogeneity. The role of biomarker-driven precision oncology and advent of immunotherapies is rapidly changing standard-of-care (SOC) treatment and improving overall survival across numerous tumour types. However, most of these approaches remain out-of-reach for patients with 'unfavourable' CUP as currently only a handful of treatments are approved irrespective of tumour type (tumour agnostic). Most targeted therapies demonstrate tumour-type dependent efficacy, exemplified by the activity of targeted inhibitors in B-RAF mutant melanoma versus inactivity in colorectal cancers[6]. Additionally, immunotherapy is increasingly indicated by biomarker presence validated by tumour type.

[1]Nucleic Acid Biomarker Team, Cancer Research UK National Biomarker Centre, The University of Manchester, Manchester, UK. [2]Division of Cancer Sciences, Faculty of Biology, Medicine and Health, The University of Manchester and The Christie NHS Foundation Trust, Manchester Academic Health Science Centre, Manchester, UK. [3]Bioinformatics and Biostatistics Team, Cancer Research UK National Biomarker Centre, The University of Manchester, Manchester, UK. [4]The Christie NHS Foundation Trust, Manchester, UK. [5]These authors contributed equally: Alicia-Marie Conway, Simon P. Pearce, Alexandra Clipson. ✉e-mail: caroline.dive@cruk.manchester.ac.uk; natalie.cook17@nhs.net; dominic.rothwell@cruk.manchester.ac.uk

Molecular characterisation approaches to predict CUP Tissue-of-Origin (TOO) are considered a gateway to better treatment stratification although debate remains as to whether treatment based on TOO predictions improves outcomes. Two large prospective randomised controlled trials reported to date have showed no improvement in overall survival in CUP cohorts as a whole using gene expression based molecular profiling from tumour tissue[7,8]. These trials are hampered by the heterogeneity and diversity of tumour types predicted and sub-optimal treatments for many of the cancer types at the time of recruitment. However, there is evidence in patients with tumours predicted to be more responsive to available therapies (for example breast, colorectal, renal clear cell, ovarian or melanoma) this approach improves survival[9,10] and this is likely to increase over time as more novel immunotherapy and targeted therapies are approved in the metastatic setting. Indeed, in a recent study[4] evaluating TOO predictions based on targeted Next Generation Sequencing (NGS) from tissue in a large cohort of patients with CUP ($n = 978$), high confidence TOO predictions were made in 41.2% of cases. Those patients that had therapy appropriate for their prediction had improved clinical outcomes and TOO predictions increased the likelihood of patient receiving targeted therapy by >2 fold[4].

A significant challenge to performing molecular profiling in patients with CUP is the lack of good quality tumour tissue for analysis. Diagnostic tissue biopsies are often small, hampered by necrosis or limited tumour content and often degraded quality due to the formalin and fixation process. Multiple rounds of investigative immunohistochemistry staining performed to give a primary tumour diagnosis exhausts tissue, and molecular profiling can often only be undertaken with a repeat invasive biopsy. Several recent CUP based molecular profiling studies report up to 60% failure rate due to inadequate tissue quantity or quality[2,11], and other CUP trials prefer or mandate repeat fresh frozen biopsies for this reason[7] (CUPCOMP trial NCT: NCT04750109, SUPERNEXT study, Australia) but this is not without potential harm to patients as well as introducing a delay in performing the biopsy. To overcome this, we developed a liquid biopsy approach with potential to combine TOO predictions with mutation analysis to stratify patients within a swifter turnaround time. We previously described a robust and sensitive genome-wide circulating-free DNA (cfDNA) methylation profiling workflow (T7-MBD-seq) that detects circulating tumour DNA (ctDNA) from patients with early stage small cell lung cancer and discriminates molecular subtypes[12]. In addition, several cancer early detection studies have demonstrated cfDNA methylation patterns predict TOO with high accuracy[13–16].

In this work we develop and test a highly-accurate TOO classifier derived from methylation profiles and apply it to a pilot CUP cohort in this proof-of-concept study. We demonstrate potential utility from a single blood draw of cfDNA analysis to facilitate diagnosis and treatment stratification, combining mutation detection with TOO predictions.

## Results and discussion
### Development of a cfDNA methylation TOO classifier, CUPiD
To build a robust multi-class TOO classifier applicable to cfDNA samples, we had to address the significant challenge of high variability in ctDNA content within cfDNA (tumour fraction, TF) that results in predominant non-cancerous cfDNA dilution of tumour-specific signal, even in metastatic cancers. To overcome this without profiling thousands of cfDNA samples, we applied a bioinformatic approach to mimic cfDNA samples of varying TF[12] (Fig. 1a). Firstly, we used publicly available DNA methylation data from tumour tissue using Infinium 450 K methylation microarrays, mainly from The Cancer Genome Atlas (TCGA), representing 29 tumour classes (9,017 tumours[17], Supplementary Fig. 1a, Supplementary Data 1) and converted methylation beta-values for each array probe into estimated T7-MBD-seq read counts. Then we created in silico mixtures by mixing these estimated

counts with previously sequenced cfDNA samples from individuals without cancer (Non-Cancer Control (NCC)) ($n = 79$, Supplementary Data 2) (Fig. 1a). Each tumour class was present across a range of TFs (0.5–10%) and NCC clinical features (age, sex, ethnicity, smoking status, comorbidities) (Fig. 1a and Methods). Preliminary classifier development demonstrated the normal liver tissue component present in NCC cfDNA could erroneously result in a liver cancer class prediction, this was remedied by the addition of non-cancerous liver tissue arrays ($n = 49$) to the non-cancer class. In total 276,108 mixtures were created across 30 classes.

Tumour-specific genomic regions were calculated using differentially methylated regions (DMRs), comparing each of the 30 classes (non-mixed samples) and selecting the 250 DMRs with greatest differences between each pairwise comparison (22,179 unique regions, Supplementary Fig. 1b). A Uniform Manifold Approximation and Projection (UMAP) dimensionality reduction using these DMRs applied to 9017 tumour samples, recapitulated class clustering seen using all regions (Fig. 1b, Supplementary Fig. 1c). An ensemble classifier, termed CUPiD, was built comprising of 100 individual gradient-boosted tree sub-classifiers with each sub-classifier trained on in silico mixtures from 80% of the arrays and NCCs (Fig. 1a). The sub-classifiers were then each tested on 'held-out' data sets comprised of mixtures of the 20% of the arrays and NCCs excluded from the training set. These sub-classifiers performed accurately across the 30 classes (Supplementary Fig. 2) with a mean multi-class area under the receiver operator curve (AUROC) of 0.980 (standard deviation 0.00521, Supplementary Fig. 1d). The resulting ensemble classifier, taking the mean prediction score across sub-classifiers for each class, had a multi-class AUROC of 0.984 (Fig. 1c). When applied to cfDNA, a tumour prediction was called when the score for a single class was >0.5, ensuring the assigned class score was higher than all remaining classes combined. An unclassified prediction was reported where all scores were <0.5 or the non-cancer class was predicted.

### CUPiD performance across multiple cancer types
We tested CUPiD on an independent test cohort of 170 cfDNA samples, including 143 cancer patient samples from 13 different tumour types, and 27 NCC samples, profiled using T7-MBD-seq (Supplementary Figs. 3a, b, Supplementary Data 3). In the cancer cfDNA samples, CUPiD predicted a correct tumour type for 121/143 patients (overall sensitivity, 84.6%), an unclassified prediction for 18/143 patients (12.6%) and an incorrect prediction for 4/143 patients (2.8%) (Fig. 2a, Supplementary Data 4). None of the 27 NCC samples were predicted as a tumour class (Fig. 2a). In the 125 samples with a tumour prediction, the correct prediction was made in 121/125 of cases (TOO accuracy, 96.8%). These results compare favourably to other cfDNA TOO classifiers in development for early cancer detection, e.g., TOO accuracy using DNA methylation was recently reported between 87.0 and 90.2% in 1393 samples with a 'cancer-like' methylation signal across 21 different tumour classes[14,16]. However, as a direct technical comparison of these different methodologies has not been made, interpretation of comparative performance should be taken with caution.

The TF for each cfDNA sample in the test cohort was estimated using ichorCNA[18] from shallow whole genome sequencing and varied by tumour type, ranging between 0–60.3% (Supplementary Fig. 3c, Supplementary Fig. 4). For 54 cancer cfDNA samples, the estimated TF was below the 3% limit of ichorCNA detection[18]. A correct prediction was made in 37 of these cases (67.9%), 16 (30%) were unclassified and one had an incorrect prediction (Fig. 2b, Supplementary Fig. 3d). This demonstrates, despite low estimated TF, methylation profiling detects tumour signal and accurately predicts TOO; only 2 samples with estimated TF > 3% were unclassified (Fig. 2b).

For the 4 cases where CUPiD predicted an incorrect tumour type, 3 were patients with lung adenocarcinoma, of which 2/3 were predicted as lung squamous carcinoma and 1/3 predicted breast cancer.

The remaining incorrect prediction was a patient with cholangio-carcinoma predicted as pancreatic adenocarcinoma. For 3/4 of these misclassified patients the errors made were within the same or adjacent tissues, and we plan to overcome this with further optimisation of the classifier to improve discrimination between tumours arising from the same or neighbouring tissues, focusing on tumour types commonly predicted in CUP cohorts.

### Determining TOO from cfDNA in a CUP cohort

This study assessed the feasibility of combining cfDNA methylation and mutation profiling with TOO predictions in a 41 patient CUP pilot study (Supplementary Data 5). Most cases were adenocarcinomas (25/41, 61.0%) or poorly differentiated carcinomas (11/41, 26.8%). Unsurprisingly, verifying TOO predictions is challenging given the intrinsic nature of CUP. Retrospectively, we reviewed clinical data including clinical history, pathology, radiology and discussions from CUP-dedicated multi-disciplinary team (MDT) meetings, including additional investigations after initial diagnosis (Supplementary Fig. 5a, Supplementary Data 5). In 15/41 (36.6%) patients, a subsequent primary tumour diagnosis was made and patients received treatment for that tumour type ('clinically resolved'). For the patients who remained

confirmed CUP (cCUP) throughout their cancer journey ($n = 26$), 18/26 had tumour diagnoses suspected based on their clinical data; either one highly-suspected primary tumour diagnosis was made ($n = 11$, 'highly suspected') or multiple primary tumour diagnoses were suspected ($n = 7$, 'differential diagnosis'). A further 8/41 patients had no clinical or radiological clues for potential primary tumour diagnosis ('no clinical suspicion'), (Supplementary Fig. 5a).

Initially, cfDNA mutational profiles were evaluated using a comprehensive 641 gene panel (see "Methods"). In the 40/41 patients where cfDNA mutation profiling was successful (Supplementary Data 6), we found 345 mutations across 33 patients (82.5% with at least one mutation, Supplementary Data 7). Per patient, the median number of mutations was 5 (range 0–77) and median variant allele frequency (VAF) was 10.4% (range 0–61.3%) (Supplementary Fig. 5b). OncoKB analysis predicted 60 alterations across 26 patients as oncogenic or likely oncogenic (Supplementary Fig. 5b) with 7/26 patients harbouring alterations that are potentially actionable with level 3 or above evidence[19]. This included 3 patients with PIK3CA mutations, 1 non-V600E BRAF mutation and 1 IDH1 mutation, all targetable with FDA approved drugs in specified tumour types (Supplementary Data 7).

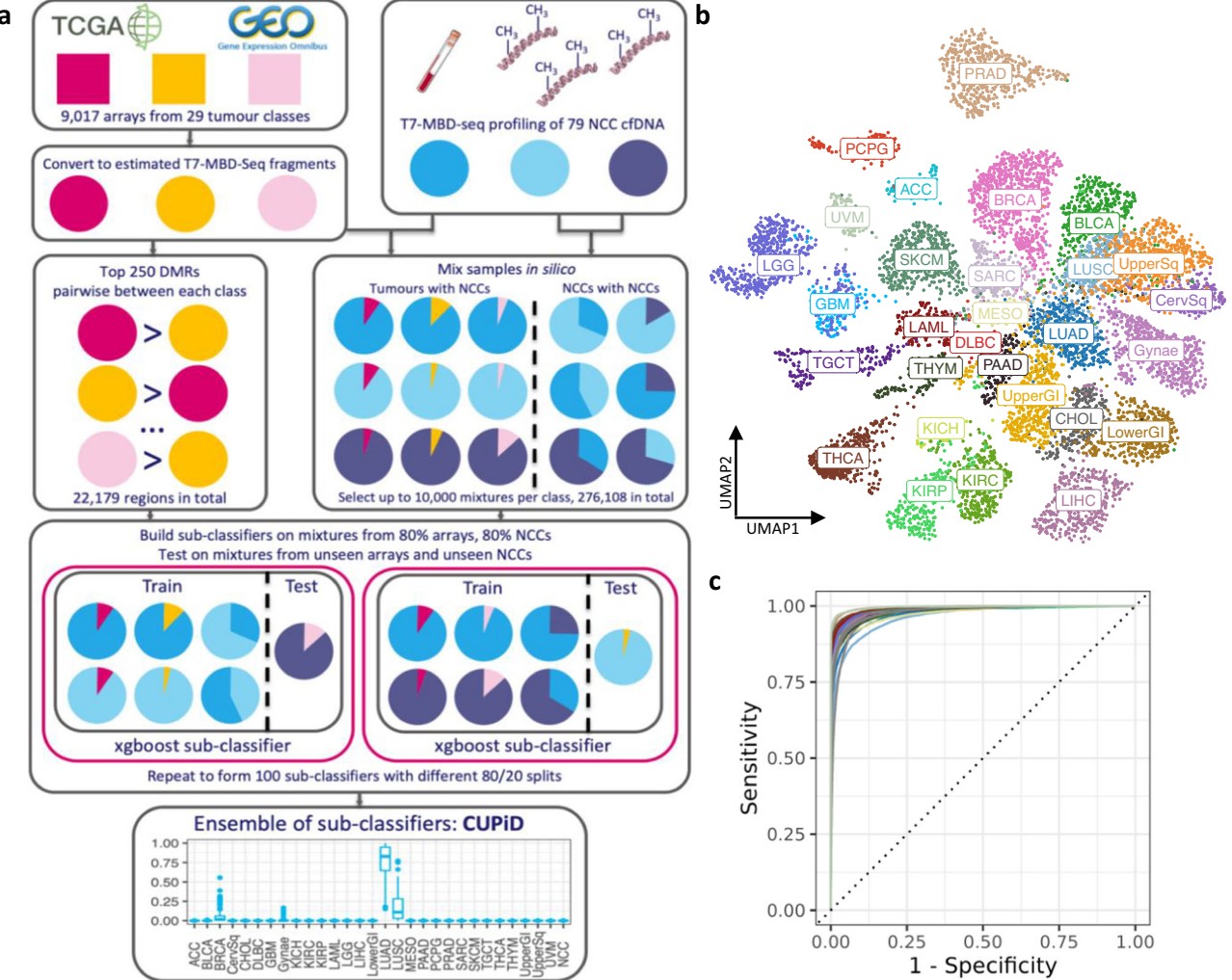

**Fig. 1 | CUPiD, an accurate tissue-of-origin classifier applicable to cfDNA.**
**a** Schematic of CUPiD development (DMR=Differentially Methylated Region).
**b** Two-dimensional Uniform Manifold Approximation and Projection for Dimension Reduction (UMAP) using 22,179 DMRs selected by tumour class by class comparison, applied to 9,017 converted methylation arrays. Class labels superimposed over centroid of members of that class. **c** Per-class Receiver Operator

Characteristic (ROC) curves for CUPiD, each using 276,108 mixture sets. Tissue-of-origin predictions for each mixture set are averaged over predictions made by those sub-classifiers not using that mixture set for training. Colours represent 30 individual classes as in Fig. 1b. Class abbreviations are defined in Table 1. Source Data are provided as a Source Data file.

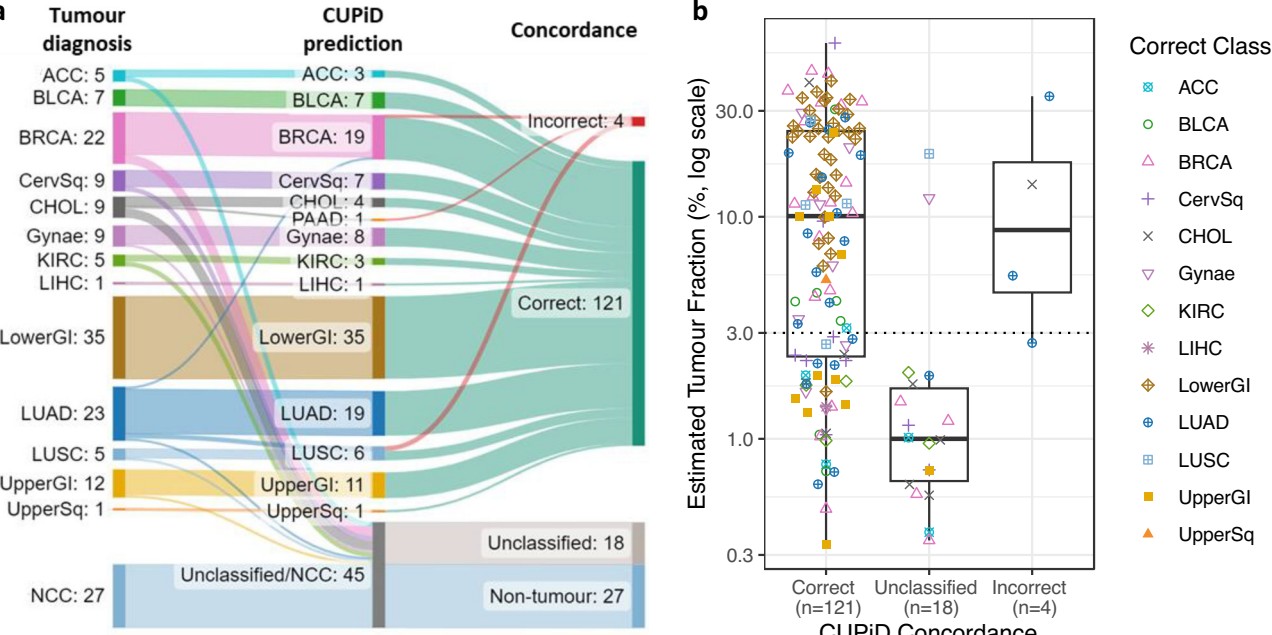

**Fig. 2 | CUPiD performance testing in cfDNA from cohort of patients with known tumour types. a** Alluvial plot showing CUPiD performance in test cohort (170 cfDNA samples; 143 known primary tumour types, 27 NCCs). **b** Estimated Tumour Fraction (TF) of 143 cfDNA samples from known primary tumour types, grouped by concordance of CUPiD predictions, coloured by correct class. Dotted line denotes limit of detection of ichorCNA (3%). Boxes mark the 25th percentile (bottom), median (central bar) and 75th percentile (top); whiskers extend to 1.5 times the interquartile range. Class abbreviations are defined in Table 1. Source Data are provided as a Source Data file.

To determine if any identified alterations could inform tumour type in the CUP cohort, we considered gene alterations previously described as significantly enriched in particular tumour types[1] and asked whether any of these tumour type-enriched alterations (TTEAs) supported the clinical features (see Methods). In no cases were TTEAs found that were supportive of only one tumour type, showing the limitations of mutation analysis. In 14/40 cases (35.0%) at least one TTEA was present and supportive (Fig. 3a and Supplementary Data 5). In 14 (35.0%) cases TTEAs were present but not supportive, or they were ambiguously supportive of several tumour types in cases of differential diagnosis. The remaining cases had no TTEAs (*n* = 12) (Fig. 3a).

cfDNA methylation profiling was successful across our entire cohort (41/41 passed QC, Supplementary Fig. 6a, Supplementary Data 8). The estimated ichorCNA TF ranged between 0 and 53.4% with 27/41 samples (65.8%) >3%, correlating well with the median VAF calculated through mutation analysis (r = 0.73, *p* < 0.00001; Supplementary Figs. 6b, c). When applied to this cohort, CUPiD yielded a tumour prediction in 32/41 of cases (78.0%) (Fig. 3a, Supplementary Data 9). For 9/41 patients (22.0%), an unclassified prediction was made, with 7/9 patients demonstrating no copy number changes (estimated TF < 3%) and 5/7 patients also had no detectable mutations suggesting low ctDNA content (Supplementary Fig. 6d). Of the 32 tumour predictions, 26 patients (81.3%) had predicted classes within 5 broad cancer categories (Fig. 3b). The most common predictions were hepato-pancreatobiliary (7/32, 21.9%, with 6 predicted cholangio-carcinomas) and female genital tract (6/32, 18.75%) (Fig. 3b). Although cholangiocarcinoma is rare, it is increasingly recognised within CUP cohorts and diagnostic biomarkers are limited[2,20,21]. Upper and lower gastrointestinal (4/32, 12.5%), lung (5/32, 15.6%) and urological cancers (4/32, 12.5%) were also frequently predicted by CUPiD. These tumour type predictions are consistent with historical data of primary tumour types found at autopsy in patients with CUP[22,23] and are commonly predicted in other large TOO studies based on tumour tissue profiling[1,7–10,24–26]. Interestingly, all of the CUPiD predicted tumour types have radically different treatment strategies compared to SOC

chemotherapy for CUP and almost all predictions would warrant consideration for immunotherapy or targeted therapies as SOC treatment options in first- or second-line setting, exemplifying the potential of our approach.

For the 33 patients that were 'clinically resolved' (*n* = 15) or suspected diagnoses (*n* = 18), 26 had a CUPiD tumour type prediction and 23/26 (88.5%) of these predictions aligned with confirmed primary tumour type or one of the suspected diagnoses ('Clinically consistent', Fig. 3a, c). Three CUPiD predictions were inconsistent with clinical data and are termed misclassified, of which 2 were 'clinically resolved' cases: the first, predicted as bladder cancer, was a rare yolk sac tumour not existing within the training data of the CUPiD classifier; the second, predicted as cholangiocarcioma, was a patient eventually determined to have gastric cancer; the final patient, predicted as an upper GI malignancy, remained as cCUP with clinical features suspicious of a breast primary (Fig. 3c). CUPiD predictions were also made in 6/8 patients where no primary was suspected or confirmed, exemplifying the potential of TOO molecular profiling in the most uncertain cases.

**Potential clinical utility for CUPiD**

The 15 patients with 'clinically resolved' primary tumours suffered a protracted period of diagnostic uncertainty and most had treatment with suboptimal empiric chemotherapy prior to primary tumour determination (Fig. 4). The median time to diagnosis for this cohort of patients was 7.1 months (range 0.4–47.2 months) and an invasive repeat biopsy was performed in 6 patients to make a final diagnosis. With adequate TF, a liquid biopsy-based TOO classifier measured at suspected cancer diagnosis, even before tissue biopsy, could have dramatically accelerated a confirmed diagnosis in a large proportion of patients and potentially negated need for repeat, invasive biopsies. In addition, precious tumour tissue material can be reserved for potential further tissue-based biomarker testing; currently a requirement for initiation of most targeted therapies and some immunotherapies. The experimental turnaround time for CUPiD is currently 3 weeks with potential to reduce via assay optimisation.

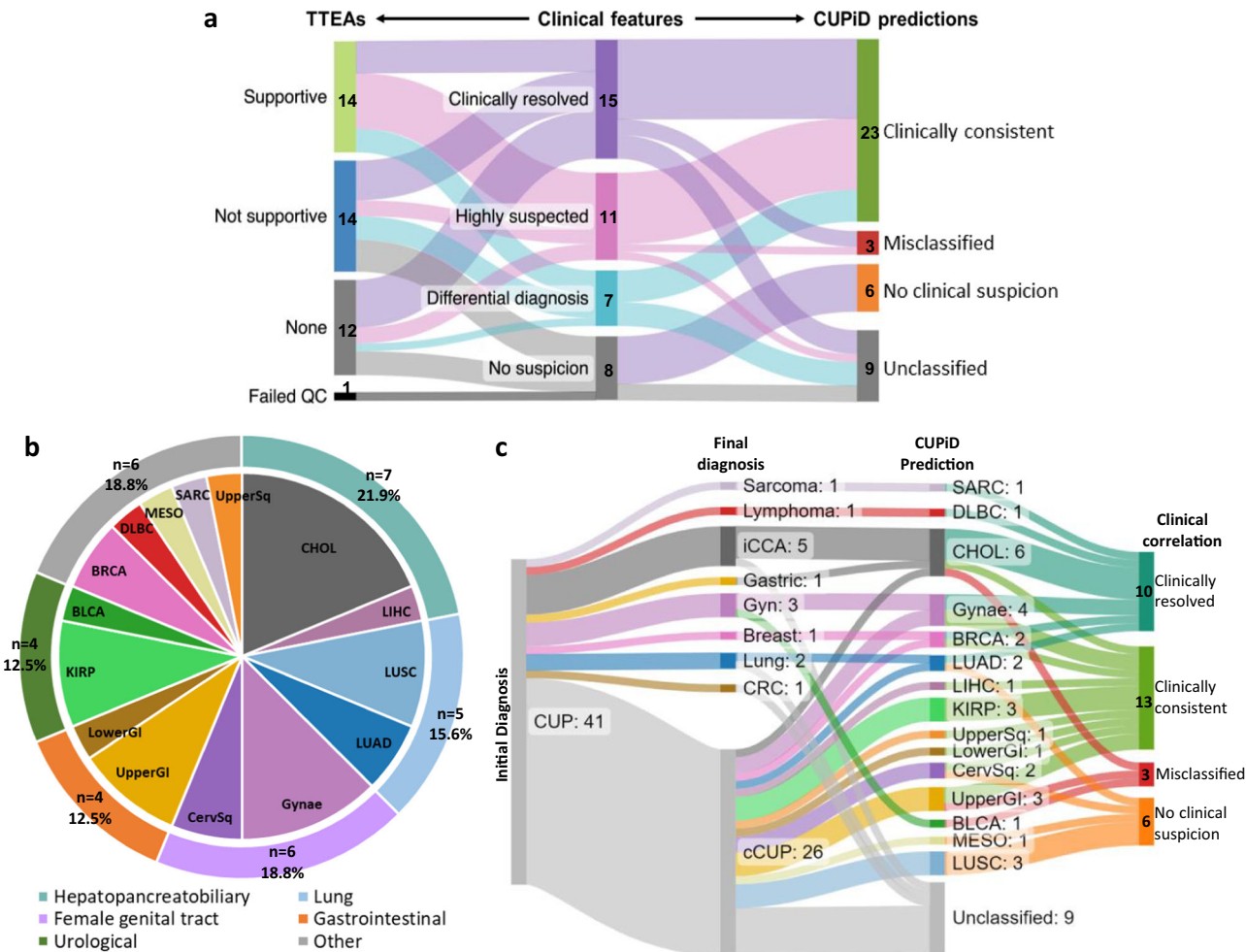

**Fig. 3 | Application of CUPiD to cfDNA from 41 patients with CUP. a** Alluvial plot showing how the tumour type-enriched alterations (TTEAs) (left) and CUPiD predictions (right) correspond to clinical classifications (centre) (QC=Quality control). **b** Distribution of tumour types predicted by CUPiD (n = 32), excluding unclassified predictions. **c** Alluvial plot showing CUPiD predictions and correlation with subsequent primary tumour diagnosis or clinical suspicion (iCCA intrahepatic cholangiocarcinoma, Gyn gynaecological cancer, CRC colorectal cancer, cCUP confirmed Cancer of Unknown Primary). Class abbreviations are defined in Table 1. Source Data are provided as a Source Data file.

In summary, we have developed CUPiD, an accurate TOO liquid biopsy with encouraging sensitivity and accuracy in known tumour types and clinically consistent predictions for patients with CUP whose cfDNA contains adequate TF. As cfDNA mutation and methylation profiling can be performed from the same blood draw, this approach enables identification of potentially actionable alterations alongside TOO predictions to aid treatment stratification. Applying CUPiD to a pilot cohort of CUP patients resulted in predicted tumour types in 32/41, with all 32 patients having the potential to benefit from radically different, tumour-specific treatment strategies compared to CUP SOC chemotherapy. Next steps are further validation of CUPiD in larger cohorts of known tumour cfDNA samples and in a statistically powered prospective CUP clinical trial.

## Methods

### Patient recruitment and sample collection

Patients with cancer were recruited through the TARGET (Tumour Characterisation to Guide Experimental Targeted Therapy) trial. Ethical approval obtained from the North-West (Preston) National Research Ethics Service in February 2015 (reference 15/NW/0078) and the trial was registered on the NIHR Central Portfolio Management System (reference CPMS ID 39172). Additional patients with CUP were recruited via the Manchester Cancer Research Centre (MCRC) Biobank

CUP Project (application number 18_ALCO_01); ethically approved through the MCRC Biobank Research Tissue Bank Ethics (ref: 07/H1003/161 + 5, ethics code 18/NW/0092). All patients were recruited at The Christie NHS Foundation Trust, a UK tertiary cancer centre.

Non-cancer-control (NCC) samples were collected, with informed consent, from three sources: 1. The Community Lung Health Study (ethically approved study London – West London & GTAC Research Ethics Committee REC reference: 17/LO415); 2. The University of Manchester healthy normal volunteer study (University of Manchester Research Ethics Committee 4 (UREC4) approval no. 2017-2761-4606); or 3. Purchased from Cambridge Bioscience (University of Manchester Research Ethics Committee approval no. 2019-7920-11797).

All patients and individuals provided fully informed written consent and research was undertaken according to Good Clinical Practice guidelines and in accordance with declaration of Helsinki.

### CUP clinical, radiological, and pathological review and data collection

Clinical data for the entire CUP cohort was retrospectively obtained and anonymised. All patients were discussed at a CUP dedicated Multi-Disciplinary Team (MDT) meeting within a tertiary cancer centre (The Christie NHS Foundation Trust) in accordance with National Institute for Health and Care Excellence (NICE) guidelines[27]. Patients therefore

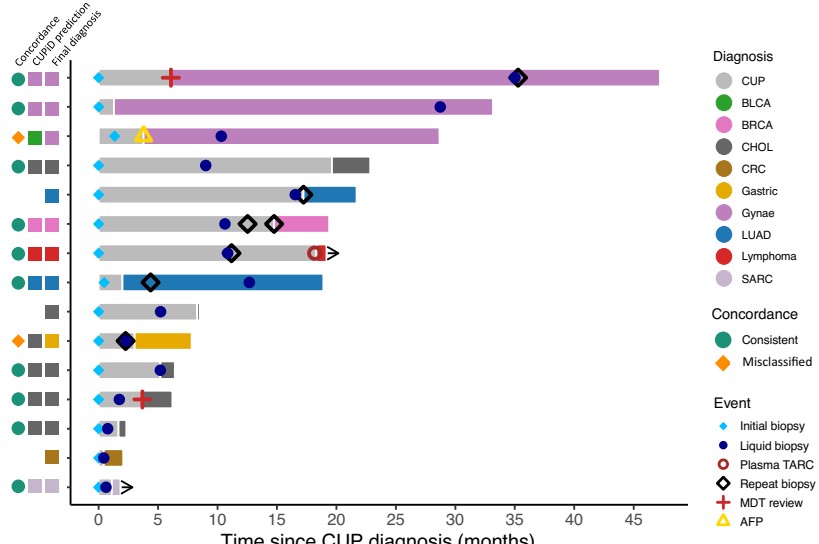

**Fig. 4 | Swimmers plot of the 15 'clinically resolved' patients.** Timeline of diagnostic investigations from point of CUP diagnosis to death or data lock. Time of final primary tumour diagnosis colour coded. Annotated by final diagnosis, CUPiD prediction and concordance. MDT multidisciplinary team meeting, AFP alpha fetoprotein, TARC thymus and activation-regulated chemokine. Class abbreviations are defined in Table 1. Source Data are provided as a Source Data file.

had histopathological and radiological review to confirm the initial diagnosis of CUP. All clinical, pathological, and radiological investigations were reviewed alongside any MDT meetings. Subsequent primary tumour diagnosis status was determined as follows: 'clinically resolved' tumour diagnosis was determined if at any point from referral to death or data lock (August 2022) a primary tumour diagnosis was documented and the patient had subsequent 'site-specific' treatment based on that diagnosis. A 'highly suspected' tumour diagnosis was categorised for patients where a single primary tumour diagnosis was suspected either prospectively or retrospectively, but patient was not treated based on this suspicion as uncertainty remained. A 'differential diagnosis' category comprised patients where clinicopathological features could narrow potential diagnoses to two or more different tumour types, with all remaining patients categorised as 'no clinical suspicion'.

### Blood sample collection
Blood samples were collected in up to 4 × 10 mL Cell-Free DNA BCT tubes (Streck, Omaha, NE), for cfDNA analysis. Plasma was separated from whole blood within 96 h of blood draw by two sequential centrifugations (2,000 g, 10 min) and stored at −80 °C before cfDNA processing. In addition, 1 × 10 mL BD Vacutainer K2 ethylenediaminetetraacetic acid (K2EDTA) sample was taken for germline DNA extraction according to study protocols.

### Circulating cell-free DNA extraction and quantification
cfDNA was isolated from up to 20 mL of plasma by using one of three isolation techniques according to the manufacturer's instructions: 1. QIAmp MinElute ccfDNA MIDI Kit (Qiagen, catalogue number 55284) 2. Using the QIAsymphony with the Circulating DNA Kit (Qiagen, catalogue number 1091063); or 3. QIAmp Circulating Nucleic Acid Kit (Qiagen, catalogue number 55154). cfDNA yields were quantified by using the TaqMan Rnase P Detection Reagents Kit (Life Technologies, catalogue number 4316831) according to manufacturer's instructions.

### Germline DNA extraction
Germline DNA was isolated from whole blood samples using the QIAmp Blood Mini Kit (Qiagen, catalogue number 51104) and sheared to 200–300 bp on the Bioruptor® Pico device (Diagenode). Sheared germline DNA was quantified using the TaqMan RNase P Detection Reagents Kit (Life Technologies, catalogue number 4316831).

### T7-MBD-seq library preparation and next generation sequencing (NGS)
For all cfDNA samples between 1 and 35 ng of cfDNA was processed via the T7-MBD-seq method[12]. Firstly cfDNA was end-repaired and A-tailed (New England Biolabs, NEB, catalogue no. E7595), dephosphorylated (FastAP Thermosensitive Alkaline Phosphatase, catalogue no. EF0654) and ligated (Roche, catalogue no. 07962355001) to custom oligonucleotides (Integrated DNA Technologies). Custom oligonucleotides (consisting of T7 RNA polymerase promoter sequence, Illumina read 1 sequencing primer-compatible sequence, a 10-bp sample barcode and a 6-bp unique molecular identifier (UMI)) were intially pre-annealed to form a hairpin loop (patent PCT/GB2020/050635). Ligated DNA was pooled (to a minimum of 75 ng in total) and spiked with 0.3 ng each of control methylated and unmethylated *Arabidopsis thaliana* DNA (Diagenode, catalogue no. C02040012). Ten percent of pooled ligated DNA was stored as a non-enriched control (NEC) sample, while the remaining 90% underwent methylation enrichment with the EpiMark Methylated DNA Enrichment kit (NEB, catalogue no. E2600S) following the manufacturer's instructions. Methylation enrichment efficiency was assessed by qPCR detection of methylated (recovery expected to be >20%) and unmethylated control DNA (recovery expected to be <1%) in enriched samples (methylation enriched; MeCap) relative to NEC samples. For both MeCap and NEC samples, amplified RNA was then generated by in vitro transcription (IVT) using a complementary T7 promoter oligonucleotide and T7 RNA polymerase (NEB, catalogue no. E2040S) as per manufacturer's instructions. After IVT, 1/3rd of the amplified RNA underwent single-strand ligation of an oligonucleotide adaptor which contained an Illumina read 2 sequencing primer-compatible sequence (NEB, catalogue no. M0373L). Subsequent reverse transcription (Thermo Scientific, catalogue no. 18-090-050) and indexing PCR library amplification (Roche, catalogue no. 07958897001) was then undertaken. Libraries were paired-end sequenced on an Illumina NextSeq 500 or NovaSeq 6000. For each sample both a MeCap fraction and NEC fraction were sequenced for genome-wide methylation profiling and copy number analysis, respectively.

## Bioinformatic data analysis software

Unless otherwise stated all data analysis was performed in R (v4.2.0) using RStudioWorkbench (v.1.4.1717-3) using the following R packages: BiocParallel (v1.30.3); BSgenome.Hsapiens.NCBI.GRCh38 (v1.3.1000); butcher (v0.3.1); dplyr(v1.0.0); GEOquery (v2.66.0); ggbeeswarm (v0.7.1); ggplot2 (v3.4.1); ggpubr (v0.4.0); glue (v1.6.2); hmmcopy (v1.32); ichorCNA (v0.3.2); janitor (v2.1.0); maftools (v2.14.0); mesa (v0.2.2); parsnip (v.1.0.0); pheatmap (v1.0.12); plyranges (v1.16.0); purrr (v1.0.1); qsea (v1.22.0); readr (v2.1.4); recipes(v1.0.3); Rsamtools (v2.12.0); stringr (v1.4.0); swimplot (v1.2.0); tibble (v3.1.8); tidyr (v1.2.0); uwot (v0.1.14); vcfR (v1.13.0); workflows (v1.1.0); xgboost (v1.6.0.1); yardstick (v1.0.0)

## T7-MBD-seq read alignment

A Nextflow (v22.04.5) DSL2 pipeline, built using the tools and modules provided by the nf-core community organisation[28] was used to process FASTQ files and produce QSEA objects as detailed. All reads were trimmed to have the same initial length of 91/61 basepairs (bp) for R1s/R2s respectively (including the 26 bp T7-MBD-seq construct at the start of R1), the unique molecular identifier (UMI) extracted using umitools[29] (v1.1.2) and the samples demultiplexed and adapter-trimmed using cutadapt[30] (v3.4). Reads were then aligned to the GRCh38 reference genome using bwa mem[31] (v0.7.17), deduplicated by the combination of R1 start position and UMI using umi-tools[29] (v.1.1.2) and mate quality scores assigned using samtools fixmate[32] (v1.15.1), to produce final bam files. Tools to quality check sequencing data were used throughout pipeline including: FastQC (v.0.11.9), Qualimap (v2.2.2d) and MultiQC (v.1.13).

## Methylation enrichment analysis

The QSEA package[33] (v1.16) was used to analyse bam files, with a custom R (v4.2.0) package (MESA, Methylation Enrichment Sequencing Analysis, v0.2.1, available from www.github.com/crukmi/mesa) to extend QSEA. The genome was tiled into 300 bp non-overlapping windows, with the removal of over-represented windows. Over-represented windows were identified from analysis of 168 non-enriched NCC fractions if the number of fragment counts within that window were in the top 0.1% of fragment counts, as well as adjacent windows with counts in the top 1%, following the method used by the ENCODE consortium[34]. Here and elsewhere a fragment represents the genomic position within the two paired ends of the sequencing read.

Fragments were filtered to only paired reads where either end of the pair mapped with a Mapping Quality (MAPQ) score of at least 10, had a fragment length between 90 and 1000 bp and a distance along the reference genome of at least 30 bp. Fragments were then uniquely assigned into windows according to the location of their midpoint. For use within QSEA, Copy Number Variations (CNV) were calculated for each sample from the non-enriched fraction, using HMMcopy[35] (v1.32) with base parameters over 1 Mbp windows. Normalised reads per million (NRPM) were generated using the CNV and the number of valid fragments in the sample, without applying trimmed mean of M values (TMM) normalisation. Beta-values (a scaled measure of methylation between 0 and 1) for each window in each sample were calculated within QSEA using the blind calibration method[33].

## IchorCNA

An estimate of Tumour Fraction (TF) for each sample was made using the non-enriched cfDNA fractions processed through IchorCNA[18] (v0.32). The 79 NCC cfDNA samples used in the generation of CUPiD were used as a panel of non-cancer samples and a 1Mbp window size applied without estimating subclonal populations. Estimated TF below 3% are considered below the limit of detection[18].

## Quality controls

NGSCheckMate[36] (v1.0.0) was applied to verify that all samples from the same individual matched as expected in the tool output. All four modalities for each patient (targeted sequencing of cfDNA and germline, T7-MBD-seq enriched and non-enriched fractions) were checked, where available.

To calculate the relative enrichment scores (relH) for the T7-MBD-seq samples, the method of the MEDIPS R package[37] (v1.42) was used. This calculates the total density of CGs contained within the mapped DNA positions on the reference sequence and divides by the total density of CGs across the whole genome. An additional QC metric for adequate methylation capture was calculated, termed the 'hyperstable fraction'. Using 805 windows that correspond to CpG sites shown to be consistently hypermethylated[38] in methylation array data from cancer and non-cancer samples, the fraction of these windows with a beta-value of 0.8 or above was calculated. This beta-value-based metric takes into account both the number of valid fragments and the global enrichment profile. For the validation set, samples with relH below 2.5 or hyperstable fraction below 0.4 are excluded. This process removed 5 patient samples and 3 non-cancer control samples from the test set, and none of the samples from the CUP cohort.

## Publicly available methylation array data

A pre-processed table of beta-values from the TCGA Pan-Cancer methylation array dataset was downloaded from https://tcga-pancan-atlas-hub.s3.us-east-1.amazonaws.com/download/jhu-usc.edu_PANCAN_HumanMethylation450.betaValue_whitelisted.tsv.synapse_download_5096262.xena.gz. This consists of 9,639 arrays (including 721 adjacent normal tissues) across 33 tumour types. Additional array data for the cholangiocarcinoma were obtained due to underrepresentation in the TCGA dataset. Using GEOquery[39] (v2.66.0) we downloaded additional pre-processed beta-values from the Gene Expression Omnibus[40–42] (accession numbers GSE89803, GSE32079, GSE49656), resulting in 256 cholangiocarcinoma arrays in total.

## Filtering and grouping of tumour samples

The TCGA Pan-Cancer methylation array dataset was filtered to exclude samples that were categorised as recurrent, redacted, metastatic, additional tumour or adjacent normal tissue (apart from liver, see below); for pancreatic ductal adenocarcinoma we removed 28 samples potentially misclassified[43]. This resulted in 8797 tumour samples and 49 normal liver tissue samples. Arrays from all of the 33 TCGA categories, and the additional cholangiocarcinoma samples from GEO, were re-grouped into classes distinguished by both anatomical location and histological subtype (Table 1, Supplementary Data 1). This enabled the classifier to be trained on more histologically distinct classes and combined classes with low numbers of samples but high similarity. For example, the TCGA oesophageal carcinoma class (ESCA) was separated by histological subtype: the oesophagus adenocarcinoma ($n = 88$) samples were combined into an upper gastro-intestinal (UpperGI) class with the stomach adenocarcinomas (STAD, $n = 393$), while the oesophagus squamous cell carcinoma ($n = 96$) was combined with the head and neck squamous cell carcinomas (HNSC, $n = 525$) to form an upper squamous class (UpperSq). Where mixed histological tumour types existed within a class, these were excluded from the training: the seven liver hepatocellular carcinoma arrays marked as "Hepatocholangiocarcinoma (Mixed)" were excluded, as were the seven cervical/endocervical arrays denoted as "Adenosquamous" (detailed in Supplementary Data 1). Table 1 summarises the classes and tumour subtypes grouped within them. This process resulted in 30 total classes, including a non-cancer class, which was included to give the classifier building process a neutral class to assign samples with low tumour fraction, rather than forcing a potentially incorrect prediction.

We then converted methylation array probe level data (beta-values) to window-based read format compatible with T7-MBD-seq read-based

**Table 1 | List of classes used within CUPiD, with the number of methylation arrays for each class and regrouping**

| CUPiD class | Class details | Total number | Source (TCGA abbreviation or GEO accession) | Number |
|---|---|---|---|---|
| ACC | Adrenocortical carcinoma | 79 | TCGA (ACC) | 79 |
| BLCA | Bladder urothelial carcinoma | 409 | TCGA (BLCA) | 409 |
| BRCA | Breast invasive carcinoma | 777 | TCGA (BRCA) | 777 |
| CervSq | Cervical squamous cell carcinoma | 254 | TCGA (CESC) | 254 |
| CHOL | Cholangiocarcinoma | 256 | TCGA (CHOL) | 36 |
| | | | GEO (GSE32079) | 50 |
| | | | GEO (GSE49656) | 32 |
| | | | GEO (GSE89803) | 138 |
| DLBC | Diffuse large B-cell lymphoma | 48 | TCGA (DLBC) | 48 |
| GBM | Glioblastoma multiforme | 139 | TCGA (GBM) | 139 |
| Gynae | Non-squamous gynaelogical carcinomas (endocervical adenocarcinoma, ovarian cystadenocarcinoma, endometrial carcinoma, uterine carcinosarcomas) | 531 | TCGA (CESC) | 46 |
| | | | TCGA (OV) | 10 |
| | | | TCGA (UCEC) | 418 |
| | | | TCGA (UCS) | 57 |
| KICH | Kidney chromophobe | 65 | TCGA (KICH) | 65 |
| KIRC | Kidney renal clear cell carcinoma | 312 | TCGA (KIRC) | 312 |
| KIRP | Kidney renal papillary cell carcinoma | 271 | TCGA (KIRP) | 271 |
| LAML | Acute myeloid leukaemia | 194 | TCGA (LAML) | 194 |
| LGG | Brain lower grade glioma | 514 | TCGA (LGG) | 514 |
| LIHC | Liver hepatocellular carcinoma | 366 | TCGA (LIHC) | 366 |
| LowerGI | Colon and rectum adenocarcinomas | 382 | TCGA (COAD) | 288 |
| | | | TCGA (READ) | 94 |
| LUAD | Lung adenocarcinoma | 456 | TCGA (LUAD) | 456 |
| LUSC | Lung squamous cell carcinoma | 364 | TCGA (LUSC) | 364 |
| MESO | Mesothelioma | 87 | TCGA (MESO) | 87 |
| PAAD | Pancreatic adenocarcinoma | 156 | TCGA (PAAD) | 156 |
| PCPG | Pheochromocytoma and Paraganglioma | 178 | TCGA (PCPG) | 178 |
| PRAD | Prostate adenocarcinoma | 495 | TCGA (PRAD) | 495 |
| SARC | Sarcoma | 257 | TCGA (SARC) | 257 |
| SKCM | Skin cutaneous melanoma | 472 | TCGA (SKCM) | 472 |
| TGCT | Testicular germ cell tumours | 149 | TCGA (TGCT) | 149 |
| THCA | Thyroid carcinoma | 503 | TCGA (THCA) | 503 |
| THYM | Thymoma | 124 | TCGA (THYM) | 124 |
| UpperGI | Stomach and oesophageal adenocarcinoma | 481 | TCGA (ESCA) | 88 |
| | | | TCGA (STAD) | 393 |
| UpperSq | Head and neck and oesophageal squamous cell carcinoma | 618 | TCGA (ESCA) | 95 |
| | | | TCGA (HNSC) | 523 |
| UVM | Uveal melanoma | 80 | TCGA (UVM) | 80 |
| NCC | Non-cancer controls (patient matched adjacent normal liver) | 49 | TCGA (LIHC) | 49 |

All array samples come from The Cancer Genome Atlas (TCGA) or Gene Expression Omnibus (GEO). Further details are included in Supplementary Data 1.

enrichment sequencing as previously described[12]. First, all 79 NCC cfDNA samples were combined to calculate an average enrichment profile typical of our T7-MBD-seq method, to generate a lookup table that determines how many counts are required for each beta-value and each CG density and can be used to convert from array beta-values to estimated T7-MBD-seq counts. The maximum beta-value was used where multiple probes lie within a single window. The resulting qseaSet estimates the counts we might expect from performing T7-MBD-seq on the sample, in the windows overlapping the array probes.

**Uniform manifold approximation and projection**
A Uniform manifold approximation and projection (UMAP[44]) was calculated on the converted tumour arrays, using the uwot R package[45] (v0.1.14) with parameters n_neighbors = 15, min_dist = 1.

**Classifier building**
To train the classifier, in silico synthetic mixture qseaSets were generated by mixing processed fragment counts between samples, either between an array sample (converted into qseaSets as detailed above) with an NCC cfDNA T7-MBD-seq sample at proportions between 0.5 and 10%, or between mixtures of two NCC T7-MBD-seq samples at proportions between 15 and 50%, all at varying numbers of fragments (between 1 million and either the number of NCC fragments or 10 million, whichever was lower). This was performed repeatedly, mixing every array sample (or NCC) once with each NCC, at a random proportion and random number of fragments. Early iterations of classifier development revealed some NCC mixtures were predicted to be liver cancer. It was hypothesised this was due to potential high levels of normal liver tissue signal within the cfDNA component for these

samples. To overcome this, we included 49 adjacent normal liver arrays from TCGA into the NCC mixture sets for training, these mixtures were assigned the NCC class.

Differentially Methylated Regions (DMRs) were calculated pairwise between classes using QSEA[33] on the converted array qseaSets (at 100% tumour fraction), and a false discovery rate (FDR) of 0.001 was applied. The difference between the average beta-values for each pair of classes, $\Delta\beta$, was calculated and used to sort DMRs by effect magnitude. The top and bottom 250 DMRs between each pairwise comparison were selected. After reduction of DMRs occurring in multiple comparisons, 22,179 distinct regions were taken forward in classifier development. For each class, we sampled up to 10,000 mixture qseaSets for classifier training, resulting in 276,108 unique samples in total, resulting in an approximately equal class distribution (3460–10,000 samples per group, depending on the number of arrays available). For each sample, the number of normalised reads per million (NRPM) were calculated for each of the 22,179 regions; this was used as the input data for each sub-classifier.

An ensemble comprising of 100 sub-classifiers was then built, with each individual sub-classifier including only the mixtures built from 80% of the array samples and 80% of the NCC samples. These sub-classifiers were built using Extreme Gradient Boosting Trees[46] (xgboost R package, v1.6.0.1) within the R tidymodels[47] (v0.2.0) framework, with default parameters except for trees = 200, sample_size = 0.5, mtry = 2135. These parameter choices resulted in 200 sequential trees built, with each tree using a randomly selected 50% of the mixtures (within the 80% stratification) and 10% of the regions. These parameters provide a large amount of variation across the population of trees in each sub-classifier, as well as the variation between sub-classifiers based on the mixtures used in training.

## Model performance

Each individual sub-classifier was tested on the remaining mixture sets comprising of samples not seen by that sub-classifier during model training (comprising 4% (20% * 20%) of the total mixture sets). Applying a sub-classifer to a held-out mixture set resulted in a prediction score for each of the 30 classes. These class-specific prediction scores for all the held-out mixture sets were compared against the ground truth to determine a one-vs-all multi-class area under the receiver operating curve (AUROC) value for each sub-classifier. To do this, we used the multiclass roc_auc function in the R package yardstick[48] (v1.0.0), which is based on the method of Hand and Till[49].

For the ensemble classifier performance, we took each individual mixture set and applied all the sub-classifiers that did not use either component of that mixture set for training. We then calculated the mean prediction score for each of the 30 classes (averaging over the relevant sub-classifer outputs). An overall multi-class AUROC value for the ensemble was again calculated using the multiclass Hand-Till method[49].

## Application of CUPiD to cfDNA samples processed through T7-MBD-seq

The ensemble of trained classifiers (CUPiD) was then applied to the 143 cfDNA samples from patients with known tumour types, 27 NCC samples not used in the generation of CUPiD and 41 samples from patients with CUP. The mean of the class-specific prediction scores across the 100 sub-classifiers was used as the final prediction score for each of the 30 classes. When applying CUPiD, a mean prediction score above 0.5 for a single class was required for a prediction of that class. This threshold ensures that the prediction value for the assigned class was higher than the remaining classes combined, as they sum to one. An unclassified prediction was reported where the mean prediction values were all <0.5 or the NCC class was predicted.

## Targeted library preparation and sequencing

cfDNA and germline DNA from all CUP samples were processed in accordance with TARGET trial laboratory protocol[50] (TARGET patients up to TAR00286) or by the following updated method (TAR00287 onward and Biobank samples). DNA repair and dA-tailing were performed using the NEBNext® Ultra™ II End Repair/dA-tailing Module (New England Biolabs, catalogue number E7546L). Adapter ligation and indexing was carried out using KAPA HyperPrep Kit (Roche, catalogue number 07962355001) with NEBNext® Multiplex Oligos for Illumina (New England Biolabs, catalogue number E7335L). Targeted NGS for whole genome libraries from cfDNA and corresponding germline DNA was carried out using SureSelect Custom DNA Target Enrichment Probes (Agilent, catalogue number 5190-4822). Target enrichment of 0.5–1.0 μg of each DNA library (paired cfDNA and gDNA libraries from the same patient pooled per pull down) was performed using SureSelect XT HS Target Enrichment System (Agilent, catalogue number G9703A) with a 641 gene hybridisation panel. Captured libraries were amplified using KAPA HiFi HotStart PCR Kits (Roche, catalogue number 07958897001) and quantified using the KAPA Library Quantification qPCR Kit (Roche, catalogue number 07960140001). Libraries were paired end sequenced at 2 ×150 bps on a NextSeq 500 or NovaSeq 6000 (Illumina).

## Targeted library alignment and mutation calling

FASTQ files were aligned to GRCh38 using bwa mem[31] (v0.7.17) and deduplicated using samtools[32] (v1.9). Mutations were called using GATK Mutect2 (v4.2.5.0, following GATK best-practices[51] with default parameters apart from a f-score beta of 5, log 10 odds threshold of 1.0 and a minimum variant allele fraction of 1%) as well as QIAGEN CLC Genomics Workbench (v20.0.2 build 200002), calling variants between the cfDNA sample and a matched germline control derived from whole blood. Those mutations called by both tools were denoted as high confidence, and annotated by VEP[52] (v193.1) and oncoKB[19] (v3.17). VCF files were converted to MAF files using vcf2maf[53] (v1.6.21), and restricted to those whose Variant_Classification field was one of Frame_Shift_Del, Frame_Shift_Ins, Splice_Site, Translation_Start_Site, Nonsense_Mutation, Nonstop_Mutation, In_Frame_Del, In_Frame_Ins or Missense_Mutation, as well as filtering out any mutations present in the gnomAD database[54] (v2.1.1) at a population frequency above 1%. The oncoplot was generated using maftools[55] (v2.8.05), selecting for those genes noted as being oncogenic by OncoKB[19].

## Mutation analysis for oncogenicity and actionability

Each high confidence genomic alteration that passed mutation filtering steps was annotated using OncoKB[19] to determine likelihood of being oncogenic and potential actionability. Only alterations actionable to Level 3 evidence and above were reported.

## Annotation of mutated genes based on enrichment for genomic alterations in individual tumour types

Genes were annotated according to tumour type enrichment using a previously described set of genes that show statistically significant enrichment for genomic alterations in individual tumour types when compared to all other cancer, based on AACR Project GENIE mutation data[1]. For patients which were 'clinically resolved' or 'highly suspected', these were considered as supported if at least one gene was noted as statistically enriched in that tumour type. For patients with a 'differential diagnosis', mutations were considered supportive if only one of the potential differential diagnoses was present; those with mutations which are enriched for more than one potential diagnosis were not considered supportive.

## Statistics and Reproducibility

Statistical analyses performed are detailed throughout the text and in figure legends with source data provided. Unless otherwise

stated, all statistical tests were two-sided and a significant result determined by a *P* value threshold of 0.05. For DMR analysis, multiple testing (FDR) correction was applied to *P* values as detailed above. For Pearson correlation hypothesis testing, data distributions were assumed to be normal but this was not formally tested. No statistical method was used to predetermine sample size and samples were chosen and processed based on the availability of plasma/cfDNA samples at the time of data generation. The experiments were not randomized. Data failing the quality control metrics were excluded as described above. In addition, NCCs with a later known cancer diagnosis were excluded. The investigators were not blinded to allocation during experiments and outcome assessment, nor to the cancer type of any of the samples. None of the NCC samples used for training and testing of classifier were used in the independent test cohort.

## Figure generation

Figure generation was performed using R (v4.2.0), using ggplot2 (v3.3.6), ggpubr (v0.4.0) and Excel (Microsoft 365). Alluvial plots were made using SankeyMATIC (https://sankeymatic.com/).

## Reporting summary

Further information on research design is available in the Nature Portfolio Reporting Summary linked to this article.

## Data availability

The T7-MBD-seq and shallow whole genome sequencing data generated in this study have been deposited under controlled access in the European Genome-Phenome Archive (EGA) with accession code EGAS00001007445. Access may be requested through the EGA request form at https://ega-archive.org/datasets/EGAD00001011178, for non-commercial purposes; a Data Access Agreement will be provided for institutional completion within two weeks.

Pre-normalised TCGA data was downloaded from Xena Browser (https://tcga-pancan-atlas-hub.s3.us-east-1.amazonaws.com/download/jhu-usc.edu_PANCAN_HumanMethylation450.betaValue_whitelisted.tsv.synapse_download_5096262.xena.gz). Previously published cholangiocarcinoma methylation arrays were downloaded from the (Gene Expression Omnibus under accession numbers GSE32079, GSE49656, GSE89803). Processed data (counts per 300 bp window) for each sample is available upon request from Zenodo (https://doi.org/10.5281/zenodo.10678015). Data from applying the classifier to each sample is available from Zenodo (https://doi.org/10.5281/zenodo.10684337). Source data are provided with this paper, except the AUROC data which is available from Zenodo due to large file size (https://doi.org/10.5281/zenodo.10684337).

## Code availability

The R package for analysis (mesa) is available from https://www.github.com/cruk-mi/mesa. The Nextflow pipeline and classifier building code is available upon request from Zenodo (https://doi.org/10.5281/zenodo.10678015). Code to analyse the output of the classifiers and generate all figures is available from Zenodo (https://doi.org/10.5281/zenodo.10684337).

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

## Acknowledgements

We sincerely thank the patients and their families for donating blood samples for research. We thank the CRUK-MI Scientific Computing team and Molecular Biology Core Facility, Manchester Cancer Research Centre Biobank, all members of the TARGET research team and members of NAB for their help and support with this project. The work was funded by Cancer Research UK (CRUK) via core-funding to the CRUK Manchester Institute (grant no. C5759/A27412; CD) and the CRUK Manchester Centre (grant no. A25254; CD), and supported by the CRUK Manchester Experimental Cancer Medicines Centre (grant no. A20465; CD), the CRUK Lung Cancer Centre of Excellence (grant no. A25146; CD), the Manchester Experimental Cancer Medicine Centre and the NIHR Manchester Biomedical Research Centre (NIHR203308, CD, NC, MGK, AMC). The work was additionally funded by The CUP Foundation: Jo's friends (AC) and the TARGET trial was funded by The Christie Charitable fund (MGK). The views expressed are those of the author(s) and not necessarily those of the funders, the NHS, the NIHR or the Department of Health. This manuscript is dedicated to Patrick P. Dive, who sadly passed away in December 2022 with CUP.

## Author contributions

A-M.C. designed and performed experiments, sourced funding, analysed data and wrote the manuscript. S.P.P. performed computational analyses, developed the classifier and wrote the manuscript. S.P.P, K.K. and S.M.H. developed a bioinformatics pipeline for T7-MBD-seq and an R package for analysis under supervision of A.K. A.C. coordinated and oversaw experiments and edited the manuscript. F.C., A-M.C., D.J.W. performed T7-MBD-seq experiments and analysis. S.M.H provided statistical advice, edited the manuscript and reviewed the classifier code. A S M M.H., S.F. and D.S.T developed a bioinformatics pipeline for cfDNA mutation analysis. C.M. provided CUP clinical input, sample collection and approved manuscript. G.B. provided initial input and support to this study. M.G.K contributed to sample collection and approved the manuscript. N.C directed research, contributed to sample collection, sourced funding, provided clinical input and edited the manuscript. C.D and D.G.R. directed the research and evolved the manuscript to the final draft.

## Competing interests

C.D. receives research grants/support from AstraZeneca, Astex Pharmaceuticals, Bioven, Amgen, Carrick Therapeutics, Merck AG, Taiho Oncology, GSK, Bayer, Boehringer Ingelheim, Roche, BMS, Novartis,

Celgene, Epigene Therapeutics Inc, Angle PLC, Menarini, Clearbridge Biomedics, Thermo Fisher Scientific, Neomed Therapeutics. C.D. has received/receives honoraria/consultancy fees from Biocartis, Merck, AstraZeneca and GRAIL. Outside of the scope of work, research funding/educational research grants has been received from by the Experimental Cancer Medicine Team (PI: Cook) from AstraZeneca, Bayer, Pfizer, Orion, Taiho, Oncology, Roche, Starpharma, Eisai, RedX, UCB, Boeringher, Merck, Stemline Tarveda and Avacta. M.G.K has received consultancy/advisory board fees from Bayer, Guaradant Health, Janssen, Roche, Seattle Genetics; speakers fees from Janssen, Roche; research funding from Novartis, Roche and travel expenses from Immutep, Janseen and Roche. The remaining authors declare no competing interests.
