## [Peer Review File · Nature Communications]

A cfDNA methylation-based tissue-of-origin classifier for Cancers of Unknown PrimaryThis manuscript has been previously reviewed at another journal that is not operating a transparent peer review scheme. This document only contains reviewer comments and rebuttal letters for versions considered at *Nature Communications*.

REVIEWER COMMENTS

Reviewer #2 (Remarks to the Author):

As discussed in my earlier review, the manuscript is well written, the bioinformatic approach is sound and validation of the tissue-of-origin classifier on cfDNA samples from cancer patients demonstrated high specificity. I agree that methylation-based TOO classification from the peripheral blood is of potential interest for the clinical management of CUP patients, in particular due to limited availability or quality of tumor tissue. Direct comparison to tissue-based molecular TOO classification would have helped to evaluate CUPiD, but as I understand, suitable tumor samples from the CUP patients within the study are not available. Without direct benchmarking of CUPiD with other published cfDNA TOO classifiers any comparison needs to be taken with caution and should be phrased accordingly. Moreover, misclassified CUP cases within the study should be indicated as such and separated from clinically resolved cases.

Reviewer #4 (Remarks to the Author):

The authors investigated the challenging topic – identifying the primary site for patients with Cancers of Unknown Primary (CUP). If we can have reliable tests, especially blood-based tests for CUP, I believe it will be beneficial for the patients. However, this topic has multiple hurdles to overcome.

Comment 1:

The difficulty of trying to find the primary cancer for the patients with cancer of unknown primary is that there is no gold standard test to identify the primary cancer.

Thus, the new technology for the identification of primary site for CUP, such as current assay with CUPiD, is difficult to confirm. Clinical correlation will not be sufficient since those case are already considered to be unknown primary.

However, there is commercially available test called CancerTYPE ID that is well accepted as a tool to identify primary cancer for patients with CUP.

<https://www.hologic.com/hologic-products/tests/cancertype-id>

Can author correlate their result with CancerTYPE ID or something similar?

Comment 2:

As pointed out by the author, there are at least two randomized trial that failed to demonstrate the benefit of primary site directed therapy based on gene profiling vs. empiric chemotherapy approach among CUP patients. Hence, I worry similar tests to identify primary site for CUP may not improve the clinical outcome.

Potentially knowing the genomic may improve the overall outcome.

Minor comment:

Although authors states below in the introduction, I believe colorectal cancer is the only exception. Thus, stating targeted therapies are tumor type dependent may be overstated.

“Apart from a handful of tumour agnostic treatments, most targeted therapies demonstrate tumour-type dependent efficacy, exemplified by activity of targeted inhibitors in B-RAF mutant melanoma versus inactivity in colorectal cancers”

RESPONSE TO REVIEWERS' COMMENTS

Reviewer #2 (Remarks to the Author):

Comment: As discussed in my earlier review, the manuscript is well written, the bioinformatic approach is sound and validation of the tissue-of-origin classifier on cfDNA samples from cancer patients demonstrated high specificity. I agree that methylation-based TOO classification from the peripheral blood is of potential interest for the clinical management of CUP patients, in particular due to limited availability or quality of tumor tissue. Direct comparison to tissue-based molecular TOO classification would have helped to evaluate CUPiD, but as I understand, suitable tumor samples from the CUP patients within the study are not available.

Response: We thank the reviewer for their positive overall feedback on the manuscript and for highlighting the importance of this approach in addressing tissue scarcity. As we previously commented we agree that a direct comparison of tissue-based molecular TOO classifiers would be an interesting evaluation of CUPiD, but as discussed this is not possible due to scarcity of tissue samples. Of the 41 patients with CUP in our study, following diagnostic work-up and trial-based mutation profiling access to tissue for methylation analysis was only possible in 6 patients. Of these, 5/6 failed our methylation QC's. We have discussed the issue of tissue availability across CUP cohorts within the manuscript (lines 66 – 76) and further discuss below in reference to the initial results of the CUPCOMP study.

Comment: Without direct benchmarking of CUPiD with other published cfDNA TOO classifiers any comparison needs to be taken with caution and should be phrased accordingly.

Response: We agree that benchmarking to other TOO cfDNA classifiers would also be interesting in evaluating CUPiD, however this is also not feasible. The best described cfDNA TOO methylation assay is GRAIL's Galleri test, (Liu et al., Ann Oncol. 2020) however we do not have access to this commercial assay, and this would be beyond the scope of the ethics and patients consent agreed on this project.

We have also altered the manuscript to rephrase that a direct technical comparison between CUPiD and other cfDNA methylation approaches has not been made and therefore interpretation of comparative performance should be taken with caution (lines 126 - 128)

Comment: Moreover, misclassified CUP cases within the study should be indicated as such and separated from clinically resolved cases.

Response: In response to the reviewer's final comment, we have renamed the three CUP cases where predictions did not align with clinical data as 'misclassified'. These misclassifications are discussed in detail within the manuscript (lines 193 – 200)

Reviewer #4 (Remarks to the Author):

Comment: The authors investigated the challenging topic – identifying the primary site for patients with Cancers of Unknown Primary (CUP). If we can have reliable tests, especially blood-based tests for CUP, I believe it will be beneficial for the patients. However, this topic has multiple hurdles to overcome.

Response: We thank Reviewer #4 for their detailed review of the manuscript and acknowledgement of the challenges faced by the topic, but also the real potential of the approach for clinical benefit for patients with CUP. We have addressed further comments below:

Comment: The difficulty of trying to find the primary cancer for the patients with cancer of unknown primary is that there is no gold standard test to identify the primary cancer. Thus, the new technology for the identification of primary site for CUP, such as current assay with CUPiD, is difficult to confirm. Clinical correlation will not be sufficient since those case are already considered to be unknown primary.

Response: We agree that a fundamental issue for tumours that arise in patients with CUP is that there is no ground-truth in determining the primary tumour type. However, clinical features, in combination with mutation patterns, can reveal suspected or confirmed primary cancer types in some patients. This is exemplified by the fact that for 15 patients in the 41 CUP cohort in this study, a primary cancer was eventually determined during the patient's cancer journey. This is higher than other published CUP datasets where a latent primary becomes apparent after a CUP diagnosis is made^{1,2}, and may reflect the contribution of mutation profiling in the determination of primary cancer type.

However, there is commercially available test called CancerTYPE ID that is well accepted as a tool to identify primary cancer for patients with CUP. <https://www.hologic.com/hologic-products/tests/cancertype-id> [hologic.com]. Can author correlate their result with CancerTYPE ID or something similar?

Response: We welcome the suggestion by the reviewer to compare (or benchmark) to other tissue-based classifiers and feel this could carry value, but there are fundamental challenges to this, both for this cohort, and for wider CUP cohorts, and with gene expression profiling specifically:

1. The premise of the study design was to overcome the challenges of scarcity of tissue in CUP driven by the fact that performing tissue-based TOO molecular profiling is hampered by a lack of good quality tissue in CUP cohorts. Indeed, in our cohort most tissue was exhausted prior to the study. In the small number of patients where we did have archival tissue (n=6) the quality of nucleic acid extraction was poor. Although we feel direct tissue comparison of predictions would be informative, we unfortunately do not have access to tissue for a direct tissue comparison.
2. Scarcity of tissue in CUP cohorts is well documented by several much larger studies including up to a 30% failure rate in a large international CUP study recently reported (CUPISCO; NCT03498521)³ and 60% failure rate recently documented in a real-world dataset⁴. Additionally, the Manchester CUP researchers co-authoring this study have recently led a study to directly compare blood and tissue-based molecular profiling in CUP (CUPCOMP; NCT047501090). Although the results are still being analysed initial comparison has shown

that molecular profiling in tissue was only successful in 42% of the cohort (n=117), compared to 92% of the cohort where successful blood-based molecular profiling was performed. The main reasons for failure of tissue profiling was: a lack of usable tissue, lack of tumour content within the tissue or failure to obtain good quality DNA (Figure 1; unpublished data).

Figure 1: Number of patients where successful molecular profiling was performed in both blood and tissue in the CUPCOMP (NCT047501090) cohort (n=117)

- TOO profiling from tissue with gene expression profiling specifically is not recommended by European guidelines due to weak supporting evidence that it improves outcomes. ESMO 2023 guidelines state: 'The clinical utility of gene expression profiling to help elucidate the likely primary is not currently supported by high-level evidence.' There is therefore no 'gold standard' to compare to in CUP.
- Finally, there are inherent challenges with comparing DNA methylation approaches to gene expression profiling as CUPs often present atypically as poorly differentiated tumours with loss of the usual protein expression on IHC. Therefore, it is probable that CUP tumours exhibit atypical gene expression profiles and so gene expression based TOO classifiers may not perform well in CUP. Of note two studies directly comparing DNA-based and RNA-based TOO approaches show discrepancies^{2,5}. The Tothill group showed in a large cohort of patients with CUP, that DNA profiling was more informative than RNA for TOO classification². We therefore feel that CancerType ID specifically, although a well-accepted tool for primary tumour determination in more common tumour type, which uses a small gene expression-based panel (92 gene panel) not designed specifically for CUP, would not be an ideal comparator. .

Due to these issues we feel that benchmarking CUPiD to a tissue-based TOO classifier is not feasible or necessary for publication of this proof of principle study and hope that the reviewer agrees.

Comment: As pointed out by the author, there are at least two randomized trial that failed to demonstrate the benefit of primary site directed therapy based on gene profiling vs. empiric chemotherapy approach among CUP patients. Hence, I worry similar tests to identify primary site for CUP may not improve the clinical outcome. Potentially knowing the genomic may improve the overall outcome.

Response: As highlighted in the manuscript, determining tissue-of-origin to guide therapy remains a debated issue in patients with CUP. The limitations of the two randomised controlled trials are discussed in the manuscript and other publications⁷. Of note the two randomised controlled trials that have failed to show improvements in clinical outcomes in CUP are based on gene expression profiling and it is unclear whether this is a failure of accurate prediction by gene expression profiling (a

potentially suboptimal test, as discussed above) or that any form of molecular tissue of origin predictions are not useful in CUP.

An inherent challenge with CUP is the heterogeneity of the cohort, with many patients with CUP ultimately having poorly differentiated carcinomas that will not respond well to any therapy. It is likely that a small, but meaningful, minority of patients may benefit from this approach, for example, those patients determined to have a highly responsive tumour type (e.g. lymphoma) or solid tumour with actionable mutation or indication for immunotherapy. Given the changing landscape of metastatic cancer treatments, and the fact targeted, and immunotherapy are usually licenced and approved in a tumour-type dependent manner, determining a primary tumour of origin remains an important clinical need for patients with CUP to access potentially efficacious treatments.

Minor comment: Although authors states below in the introduction, I believe colorectal cancer is the only exception. Thus, stating targeted therapies are tumor type dependent may be overstated “Apart from a handful of tumour agnostic treatments, most targeted therapies demonstrate tumour-type dependent efficacy, exemplified by activity of targeted inhibitors in B-RAF mutant melanoma versus inactivity in colorectal cancers”

Response: We have taken on board the reviewers comments and acknowledge that there is increasing evidence of tumour-agnostic efficacy for both targeted and immunotherapy. Though it remains that the majority of therapies are licenced and approved in a tumour-type dependent manner and therefore access to these drugs is dependent on a primary tumour diagnosis. We have therefore amended the manuscript accordingly (lines 44 – 49):

“However, most of these approaches remain out-of-reach for patients with ‘unfavourable’ CUP as currently only a handful of treatments are approved irrespective of tumour type (tumour agnostic). Most targeted therapies demonstrate tumour-type dependent efficacy, exemplified by the activity of targeted inhibitors in B-RAF mutant melanoma versus inactivity in colorectal cancers⁶. Additionally, immunotherapy is increasingly indicated by biomarker presence validated by tumour type..”

1. Moran, S., *et al.* Epigenetic profiling to classify cancer of unknown primary: a multicentre, retrospective analysis. *The Lancet Oncology* **17**, 1386-1395 (2016).
2. Posner, A., *et al.* A comparison of DNA sequencing and gene expression profiling to assist tissue of origin diagnosis in cancer of unknown primary. *J Pathol* (2022).
3. Pauli, C., *et al.* A Challenging Task: Identifying Patients with Cancer of Unknown Primary (CUP) According to ESMO Guidelines: The CUPISCO Trial Experience. Vol. 26 e769-e779 (2021).
4. Huey, R.W., *et al.* Feasibility and Value of Genomic-Profiling in Cancer of Unknown Primary: Real-World Evidence from Prospective Profiling Study. *J Natl Cancer Inst* (2023).
5. Möhrmann, L., *et al.* Comprehensive genomic and epigenomic analysis in cancer of unknown primary guides molecularly-informed therapies despite heterogeneity. *Nature Communications* **13**, 4485 (2022).
6. Weiss, L.M., *et al.* Blinded Comparator Study of Immunohistochemical Analysis versus a 92-Gene Cancer Classifier in the Diagnosis of the Primary Site in Metastatic Tumors. *The Journal of Molecular Diagnostics* **15**, 263-269 (2013).
7. Conway, A.M., Mitchell, C. & Cook, N. Challenge of the unknown: How can we improve clinical outcomes in cancer of unknown primary? *Journal of Clinical Oncology* **37**(2019).

REVIEWERS' COMMENTS

Reviewer #2 (Remarks to the Author):

My comments were addressed appropriately by the authors.

Reviewer #4 (Remarks to the Author):

Thank you for answering the comments.

The real question is if we can improve overall survival by identifying the origin of cancer. This is not the aim of this study and I hope the current study will facilitate uncovering the unknown.